# Worldwide epidemiology of Crimean-Congo Hemorrhagic Fever Virus in humans, ticks and other animal species, a systematic review and meta-analysis

Jean Thierry Ebogo Belobo[1,2], Sebastien Kenmoe[3]*, Cyprien Kengne-Nde[4], Cynthia Paola Demeni Emoh[5], Arnol Bowo-Ngandji[5], Serges Tchatchouang[6], Jocelyne Noel Sowe Wobessi[3], Chris Andre Mbongue Mikangue[5], Hervé Raoul Tazokong[5], Sandrine Rachel Kingue Bebey[5], Efietngab Atembeh Noura[1], Aude Christelle Ka'e[7], Raïssa Estelle Guiamdjo Simo[5], Abdou Fatawou Modiyinji[3], Dimitri Tchami Ngongang[5], Emmanuel Che[5], Sorel Kenfack[5], Nathalie Diane Nzukui[8], Nathalie Amvongo Adjia[1], Isabelle Tatiana Babassagana[5], Gadji Mahamat[5], Donatien Serge Mbaga[5], Wilfred Fon Mbacham[2], Serge Alain Sadeuh-Mbah[3], Richard Njouom[3]*

1 Medical Research Centre, Institute of Medical Research and Medicinal Plants Studies, Yaoundé, Cameroon, 2 Department of Biochemistry, Faculty of Science, The University of Yaounde I, Yaoundé, Cameroon, 3 Virology Department, Centre Pasteur of Cameroon, Yaoundé, Cameroon, 4 Epidemiological Surveillance, Evaluation and Research Unit, National AIDS Control Committee, Yaoundé, Cameroon, 5 Department of Microbiology, Faculty of Science, The University of Yaounde I, Yaoundé, Cameroon, 6 Bacteriology Department, Centre Pasteur of Cameroon, Yaoundé, Cameroon, 7 Virology Department, Chantal Biya International Reference Centre, Yaoundé, Cameroon, 8 School of Health Sciences-Catholic University of Central Africa, Department of Medical Microbiology, Yaoundé, Cameroon

* kenmoe@pasteur-yaounde.org (SK); njouom@pasteur-yaounde.org (RN)

## Abstract

There are uncertainties about the global epidemiological data of infections due to Crimean-Congo hemorrhagic fever virus (CCHFV). We estimated the global case fatality rate (CFR) of CCHFV infections and the prevalence of CCHFV in humans, ticks and other animal species. We also explored the socio-demographic and clinical factors that influence these parameters. In this systematic review with meta–analyses we searched publications from database inception to 03rd February 2020 in Pubmed, Scopus, and Global Index Medicus. Studies included in this review provided cross-sectional data on the CFR and/or prevalence of one or more targets used for the detection of CCHFV. Two independent investigators selected studies to be included. Data extraction and risk of bias assessment were conducted independently by all authors. Data collected were analysed using a random effect meta-analysis. In all, 2345 records were found and a total of 312 articles (802 prevalence and/or CFR data) that met the inclusion criteria were retained. The overall CFR was 11.7% (95% CI = 9.1–14.5), 8.0% (95% CI = 1.0–18.9), and 4.7% (95% CI = 0.0–37.6) in humans with acute, recent, and past CCHFV infections respectively. The overall CCHFV acute infections prevalence was 22.5% (95% CI = 15.7–30.1) in humans, 2.1% (95% CI = 1.3–2.9) in ticks, and 4.5% (95% CI = 1.9–7.9) in other animal species. The overall CCHFV recent infections seroprevalence was 11.6% (95% CI = 7.9–16.4) in humans and 0.4% (95% CI = 0.0–2.9) in other animal species. The overall CCHFV past infections seroprevalence was 4.3%

**Data Availability Statement:** All relevant data are within the manuscript and its Supporting Information files.

**Funding:** The author(s) received no specific funding for this work.

**Competing interests:** The authors have declared that no competing interests exist.

(95% CI = 3.3–5.4) in humans and 12.0% (95% CI = 9.9–14.3) in other animal species. CFR was higher in low-income countries, countries in the WHO African, South-East Asia and Eastern Mediterranean regions, in adult and ambulatory patients. CCHFV detection rate in humans were higher in CCHFV suspected cases, healthcare workers, adult and hospitalized patients, ticks of the genus *Ornithodoros* and *Amblyomma* and in animals of the orders *Perissodactyla* and *Bucerotiformes*. This review highlights a significant disease burden due to CCHFV with a strong disparity according to country income levels, geographic regions, various human categories and tick and other animal species. Preventive measures in the light of these findings are expected.

## Author summary

Crimean-Congo hemorrhagic fever is one of the most severe zoonotic viral disease that occurs in humans. It is therefore necessary to provide public health stakeholders, research funding agencies and healthcare workers with accurate data on the burden of this disease in order to guide decision-making priorities. Our study is the first systematic review with meta-analysis to provide global data on CCHFV CFR in humans, CCHFV prevalence and seroprevalence in humans, ticks and other animal species. This review is also the first to maps CCHFV CRF, prevalence, and seroprevalence in humans, ticks and other animal species according to the country income level, geographic region, various human categories, and extensive tick and animal species. Broadly, the study showed elevated CFR in low-income countries, WHO regions of Africa, South-East Asia and Eastern Mediterranean, and adult and outpatient patients. In addition, the prevalence and seroprevalence of CCHFV were higher in CCHFV suspected cases, healthcare workers, adults and hospitalized patients, ticks of the genus *Ornithodoros* and *Amblyomma*, and animals of the orders *Perissodactyla* and *Bucerotiformes*. Finally, our findings show that more attention needs to be paid to low-income countries particularly in WHO regions of Africa, South-East Asia, and Eastern Mediterranean in order to prevent human deaths due to CCHFV. In particular low-income countries and adults should benefit from emergency measures aimed at improving the management of patients with CCHF and reducing exposure of humans and animals to ticks. This meta-analysis further shows that apart from the monitoring and control of CCHFV in humans, a special attention should also be given to role played by non-*Hyalomma* tick species and other animal species, both domestic and wild.

## Introduction

Crimean-Congo Hemorrhagic Fever (CCHF) is one of the most severe zoonotic viral diseases that occur in humans. This disease characterized by fever and hemorrhage, often with nonspecific signs and symptoms with case fatality rates (CFR) ranging from 5–30% [1, 2]. The Crimean-Congo hemorrhagic fever virus (CCHFV) was first identified in the Crimean region of Russia in 1944 and was subsequently shown to be identical to the Congo virus identified in the Congo basin in 1967, giving the virus its current name [2–4]. This virus has been detected in more than 50 countries of Asia, Europe, and Africa where it is associated either to outbreaks of hemorrhagic fever or only sporadic cases [2]. The presence of the ticks, CCHFV, CCHF, and death due to CCHFV are increasing in endemic areas and also in new areas [5, 6]. This is due

to various factors such as climate change, the increase in the tick number, the increasing exposure of animals and humans and the improvement of viral detection assays.

CCHF is a tick-borne disease caused by CCHFV belonging to the genus *Orthonairovirus* of the family *Nairoviridae* within the order *Bunyavirales* [7, 8]. CCHFV is maintained in nature through transmission by ticks of the family *Ixodidae*, and members of the genus *Hyalomma* are considered as the main vectors that spread the virus to humans and a variety of wild and domestic animals [9]. Both wild and domestic animals represent an important link in the disease transmission cycle, and play a key role in the amplification of the virus [10]. Animals may serve as asymptomatic reservoirs of CCHFV and their distribution to human cases appears to be closely related to vector distribution [2]. Humans are infected by tick bite, crushing it on an open wound, contact with blood, body fluids or tissues of a viremic animal or human, and possibly through sexual transmission [1, 11, 12]. Virus (re)emergence continue to be key topics of national and international health security and CCHF is classified by the World Health Organization (WHO) as a priority disease for research and development due to its potential to cause major epidemics in humans [13]. As other hemorrhagic fever viruses, appropriate knowledge about the CCHFV ecology, transmission dynamics, and competent reservoir hosts and vectors are required to anticipate the potential risk of outbreak and to better understand the disease burden of CCHF in diverse regions of the world. The surveillance of this CCHFV by assessing the status of CCHFV-specific antibodies in the animal and human populations and the presence of CCHFV in ticks are good indicators of the presence or absence of CCHFV in a given area [3, 14]. Moreover, these indicators are critical to assess the potential threat of CCHFV on human health in the epidemiological contexts that are considered.

Previous reviews and systematic reviews on CCHFV have focused on seroepidemiological studies in domestic and wild animals [10], CCHF related to travel [15], the role of vertebrate animals and ticks in CCHFV maintenance and amplification of the infection,[1, 9] and seroprevalence of CCHF in humans of the WHO European region [16]. Furthermore, Nasirian and collaborators recently performed 2 systematic reviews in the global context on the seroprevalence of CCHFV in humans and animals and the CCHFV CFR in humans [6, 17]. This systematic review with meta-analysis seeks to assess CCHFV prevalence and seroprevalence in wildlife, livestock, ticks and humans. This systematic review also provides overall data about the CCHFV CFR in various human categories. A summary of such a comprehensive and extensive literature review on CCHFV would serve as a basis to guide priorities in focusing prevention efforts according to the types of CCHFV host, different geographic areas and other multiple socio-demographic parameters.

## Methods

### Design and inclusion criteria

This systematic review and meta-analyses was performed following the guidelines of Preferred Reporting Items for Systematic Reviews and Meta-Analyses (PRISMA) [18] (S1 Table) and was registered to Prospero under the identification CRD42020167714. The methodology used for this review has been previously described [19]. Our inclusion criteria were observational or interventional studies published in peer-reviewed journals including case series, case-control, baseline data for cohort, cross-sectional studies, and epidemiological surveys. The studies included in this review provided cross-sectional data on the CFR or prevalence of one or more diagnostic targets of the CCHFV (live virus, viral antigen, viral RNA, IgM or IgG specific antibodies). We considered studies using any type of laboratory assay to find these specific CCHFV diagnostic targets in any type of sample. We classified the CCHFV infections as: acute infections (evidenced by the presence of live virus, viral antigen or RNA), recent infections

(IgM) or past infections (antibody or IgG). We defined acute infection rates as "prevalence" and recent and past infection rates as "seroprevalence". We considered all studies worldwide without any geographic restriction. We considered studies in humans, ticks, and all other domestic and wild animals considered in included studies. We included studies testing CCHFV in individual or pooled ticks. For studies testing CCHFV in individual ticks we estimated the prevalence. For studies testing pooled ticks we collected the names of positive species and considered null prevalence for negative species. We grouped and calculated the CCHFV prevalence in ticks according to their genus [20]. We grouped and calculated the CCHFV prevalences in other animals according to their orders [21]. We considered the individual CCHFV prevalences in the articles according to the type of infections (acute, recent or past) and the category of the population. A single article could thus contribute for several prevalence records. When the same population represented several prevalence data for the same type of infection, we combined the prevalence for all detection assays used or chose the technique reporting the highest prevalence. Exclusion criteria were systematic reviews, case reports, reports, commentaries, duplicate studies, and articles written in any language other than English or French.

## Data sources and search strategy

We performed a comprehensive review of electronic bibliographic databases for publications from inception till February 03, 2020. The search strategy was implemented in Pubmed (S2 Table), Scopus, and Global Index Medicus databases. We also performed manual searches in the reference section of included articles as well as relevant previous reviews for additional inclusions.

## Study selection, data extraction and assessment of study quality

All the articles identified were first reviewed based on the title and abstract by two authors (SK and JTEB), and afterwards they were reviewed based on the full text. Disagreement was resolved by discussion and consensus and if necessary, the opinion of a third reviewer was required. Data from the included studies was extracted using a Google form by 18 study authors. From the eligible articles, data extracted included: first author name, year of publication, period of study participant recruitment, study design, sampling method, timing of data collection, country, United Nations Statistics Division (UNSD) region [22], country income level [23], age range, mean or median age, male percentage, recruitment setting, hospitalization status, population category, tick genus, tick species identification approach, description of tick pooling approach, order of other animals, CCHF case definition, CCHF detection assay, CCHFV diagnostic target detected, type of infection (acute, recent or past infections), type of sample tested, the number of samples tested for CCHFV, the number of samples tested positive for CCHFV and the number of deaths among the CCHFV individuals. Data were also collected for the 10 questions for risk of bias assessment as previously described by hoy and colleagues (S3 Table) [24].

## Statistical analysis

We performed a meta-analysis using a random effect model on the data collected in the included articles to determine the CFR of CCHFV in humans and the prevalence of CCHFV in humans, ticks and other animals [25, 26]. We performed a Chi-square test to estimate the heterogeneity of the studies [25, 27]. We also determined the value of I2 which represents the proportion of the variability that could be attributable to heterogeneity and not to sampling error. We used the Egger statistical test and the funnel plot to estimate publication bias [28].

We performed a subgroup analysis to investigate the qualitative variables likely to influence the prevalence of CCHFV for each population category represented by 3 or more studies. These variables included study design, sampling method, timing of data collection (retrospective/prospective), country, country level income, UNSD region, age range, recruitment context (rural/urban), hospitalization status and type of sample. To determine the potential sources of heterogeneity, we conducted a meta-regression including subgroup analysis variables and continuous variables (mean or median age and male percentage). We conducted sensitivity analyses including only studies with low risk of bias and cross-sectional studies. Analyses were performed using R software version 3.6.2 and p values <0.05 were considered statistically significant [29, 30].

## Results

### Included studies search process

Our searches found 2345 records. We eliminated 600 duplicates and reviewed the titles and abstracts of 1,745 articles. The review of titles and abstracts eliminated 1,110 articles, reducing the number of articles requiring full-text review to 635. A total of 312 (802 prevalence and/or CFR data) articles were included in the qualitative and quantitative synthesis. Fig 1 and S4 Table illustrate the PRISMA flowchart and the specific reasons for excluding some articles after full text review.

### Assessment of study quality

None of the included studies had a high risk of bias (S5 Table). Overall (802 studies), the majority of studies were at moderate risk of bias, 680/802 (84.8%). A group of 15/802 (1.8%) studies were nationally representative, 156/802 (19.4%) had a randomized study participant recruitment, 150/802 (18.7%) reported a CCHF case definition, 17/802 (2.1%) had a participant response rate greater than 70% and 435/802 (54.2%) had a study period greater than 1 year. For studies in humans (323 studies), 203/323 (62.5%) had a moderate risk of bias, 12/323 (3.7%) studies were representative of the national population, 45/323 (13.8%) had a recruitment of randomized participants, 14/323 (4.3%) had a participant response rate greater than 70% and 178/323 (55.1%) had a study duration greater than 1 year. All studies in ticks (245) were at moderate risk of bias. Among individual ticks (209 studies), no study was representative of the national population, 73/209 (34.9%) had a randomized study participant recruitment, and 98/209 (46.8%) had a study duration greater than 1 year. For studies in pooled ticks (36 studies), none were representative of the national population, 2/36 (5.5%) had a randomized study participant recruitment, and 14/36 (38.8%) had a study duration greater than 1 year. Almost all the studies in other animals (234 studies) were at moderate risk of bias, 240/234 (99.1%). A group of 3/234 (1.2%) studies in other animals were representative of the national population, 39/234 (15.3%) had a randomised study participant recruitment and 145/234 (61.9%) had a study duration greater than 1 year.

### Baseline characteristics of included studies

The majority of studies were conducted in humans (323 studies), followed by ticks (245 studies) and other animals (234 studies) (S6 and S7 Tables). The 802 studies reporting the prevalence and/or CFR of CCHFV were published between 1974 and 2020 with participants recruited between 1964 and 2018. All studies in ticks and other animals were cross-sectional. The majority of articles were non-probabilistic, prospective and community-based. A total of 158 studies were carried out in Turkey and 126 in Iran. The most represented UNSD regions

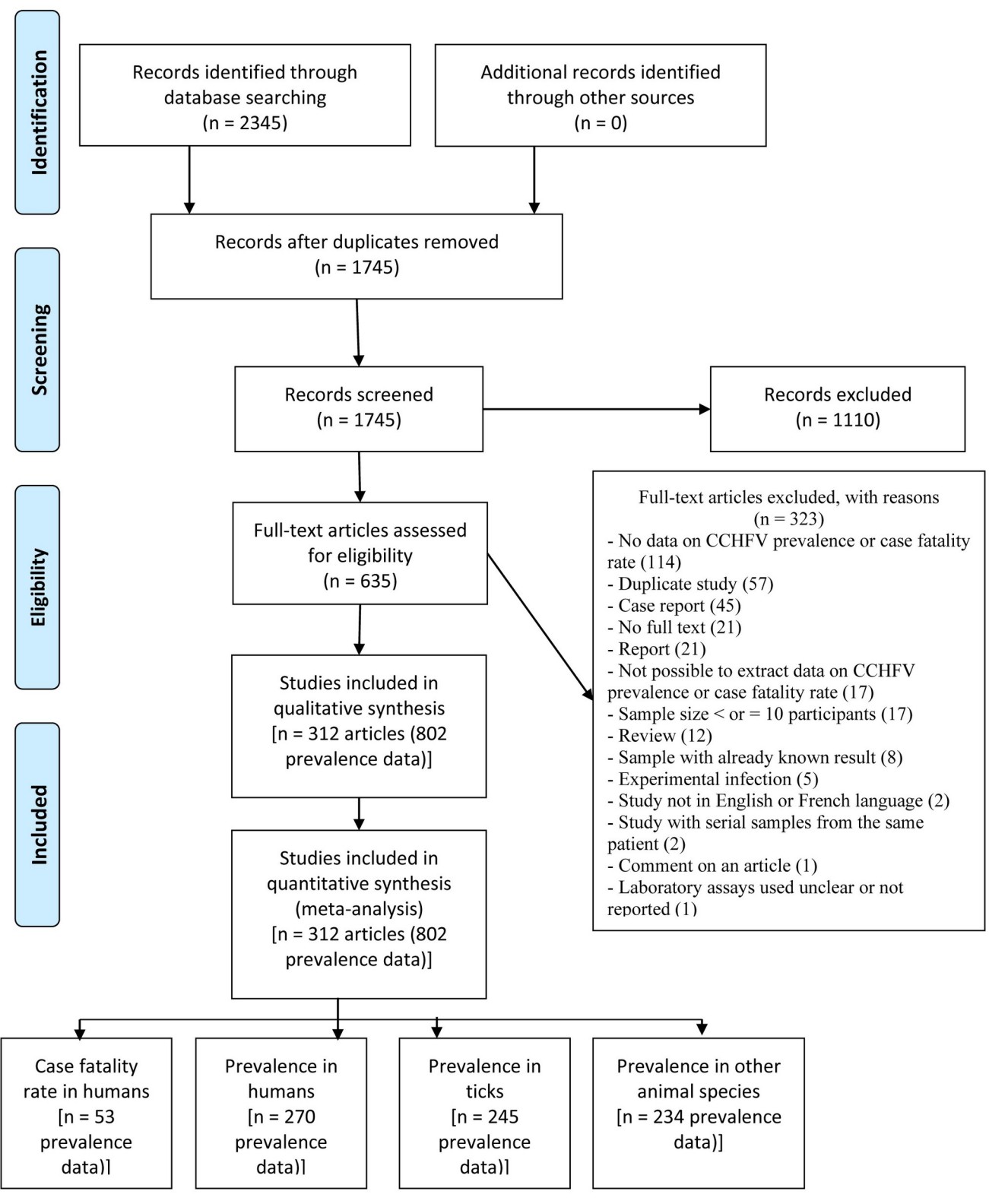

**Fig 1. Flowchart for study selection.**

were West Asia (225 studies) and South Asia (210 studies). Most of the studies were conducted in upper-middle-income countries (433 studies). Only 6 studies in individually tested ticks carried out their taxonomic assignment based on tick DNA sequence analyses. Ticks from the genus *Hyalomma* (104 studies) and *Rhipicephalus* (51 studies) were predominantly represented including the species *Hyalomma marginatum* (22 studies) and *Rhipicephalus sanguineus* (18 studies). Other animal species studied to date belong to more than 20 orders (more than 80 species). Animals of the order *Artiodactyla* were largely represented with, 157 studies, including 48 studies in sheep and 35 in goats. The studies in humans recruited mostly CCHFV suspected cases (120 studies) and febrile patients (46 studies). Thirty-six studies reported CCHFV positive tick species when determining CCHFV prevalence by pooled analyses. The predominant detection assays used to detect CCHFV in the included studies were indirect ELISA (277 studies) and conventional RT-PCR (181 studies). The majority of studies found target of past CCHFV infections evidenced by the detection of IgG antibodies (367 studies), and acute infections evidenced by the detection of viral antigen, viral RNA or live virus (367 studies). All studies in ticks involved only CCHFV acute infection. With respect to studies among humans and other animals, most of them (540 studies) found CCHFV in serum specimens.

## Results of the meta-analysis

The majority of studies on the CCHFV CFR were conducted in Turkey (35/53; 66.0%). The overall CCHFV CFR recorded in 8096 humans with acute infections recruited from 41 articles (45 reported CFR data) was 11.7% [95% CI = 9.1–14.5] (Figs 2 and S1, and S1 Text). The overall CCHFV CFR recorded in 179 humans with recent infections recruited from 6 articles (6 reported CFR data) was 8.0% [95% CI = 1.0–18.9]. The overall CCHFV CFR recorded 41 humans with past infections recruited from 2 articles (2 reported CFR data) was 4.7% [95% CI = 0.0–37.6].

The overall prevalence of CCHFV recorded in 35,198 human participants with acute infections recruited from 62 articles (67 reported prevalence data) was 22.5% [95% CI = 15.7–30.1] (Figs 3 and S2, and S2 Text). The overall seroprevalence of CCHFV recorded in 27,173 human participants with recent infections recruited from 54 articles (56 reported seroprevalence data) was 11.6% [95% CI = 7.6–16.4]. The overall seroprevalence of CCHFV recorded in 74,900 human participants with past infections recruited from 125 articles (147 reported seroprevalence data) was 4.3% [95% CI = 3.3–5.4].

All of the tick studies registered acute CCHFV infection. The overall prevalence of CCHFV recorded in 31,117 ticks tested individually or grouped (those with null prevalence) in 71 articles (209 prevalence data) was 2.1% [95% CI = 1.3–2.9] (Figs 3 and S3, and S3 Text).

The overall prevalence of CCHFV recorded in 810 other animal species with acute infections in 6 articles (10 prevalence data) was 4.5% [95% CI = 1.9–7.9] (Figs 3 and S4, and S4 Text). The overall seroprevalence of CCHFV recorded in 1,627 other animal species with recent infections in 4 articles (6 seroprevalence data) was 0.4% [95% CI = 0.0–2.9]. The overall seroprevalence of CCHFV recorded in 92,058 other animal species with past infections in 77 articles (218 prevalence data) was 12.0% [95% CI = 9.9–14.3].

In the sensitivity analysis, most of the overall results were similar to the results of cross-sectional studies and those with low risk of bias (Table 1). Compared to the overall results, the CCHFV CFR was very high in studies with low risk of bias in patients with recent and past infection. Compared to the overall results, the prevalence of CCHFV was high in studies with a low risk of bias in patients with recent infection. The overall results showed substantial heterogeneity for CCHFV CFR and CCHFV prevalence in humans, ticks, and other animal

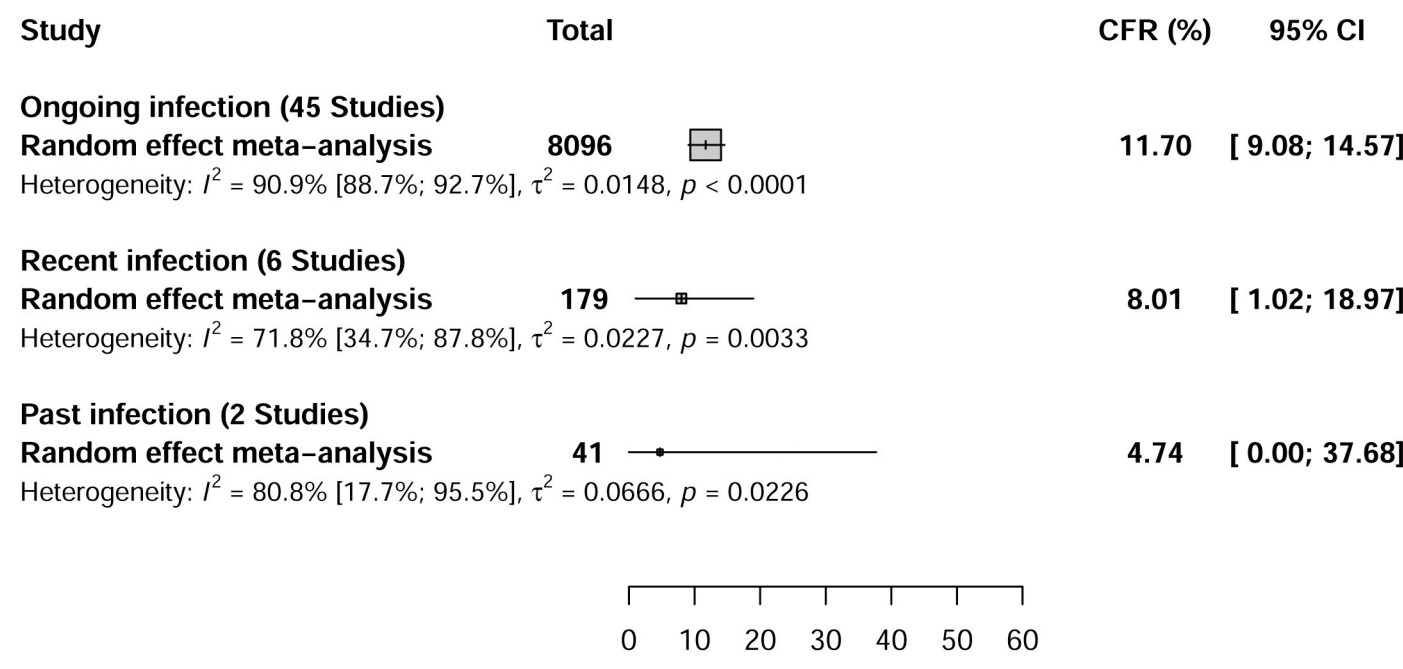

| Study | Total | | CFR (%) | 95% CI |
|---|---|---|---|---|
| **Ongoing infection (45 Studies)** | | | | |
| **Random effect meta–analysis** | **8096** | | **11.70** | **[ 9.08; 14.57]** |
| Heterogeneity: $I^2$ = 90.9% [88.7%; 92.7%], $\tau^2$ = 0.0148, $p$ < 0.0001 | | | | |
| **Recent infection (6 Studies)** | | | | |
| **Random effect meta–analysis** | **179** | | **8.01** | **[ 1.02; 18.97]** |
| Heterogeneity: $I^2$ = 71.8% [34.7%; 87.8%], $\tau^2$ = 0.0227, $p$ = 0.0033 | | | | |
| **Past infection (2 Studies)** | | | | |
| **Random effect meta–analysis** | **41** | | **4.74** | **[ 0.00; 37.68]** |
| Heterogeneity: $I^2$ = 80.8% [17.7%; 95.5%], $\tau^2$ = 0.0666, $p$ = 0.0226 | | | | |

0  10  20  30  40  50  60

**Fig 2. Global case fatality rate estimate of Crimean-Congo hemorrhagic fever virus infections in humans.**

| Study | Total | | Prevalence/Seroprevalence (%) | 95% CI |
|---|---|---|---|---|
| **Humans–Ongoing infection (67 Studies)** | | | | |
| **Random effect meta–analysis** | 35198 | | 22.54 | [15.78; 30.08] |
| Heterogeneity: $I^2$ = 99.5% [99.5%; 99.5%], $\tau^2$ = 0.1184, $p$ = 0 | | | | |
| **Humans–Recent infection (56 Studies)** | | | | |
| **Random effect meta–analysis** | 27173 | | 11.66 | [ 7.59; 16.41] |
| Heterogeneity: $I^2$ = 98.8% [98.7%; 99.0%], $\tau^2$ = 0.0597, $p$ = 0 | | | | |
| **Humans–Past infection (147 Studies)** | | | | |
| **Random effect meta–analysis** | 74900 | | 4.31 | [ 3.31; 5.42] |
| Heterogeneity: $I^2$ = 97.7% [97.5%; 97.9%], $\tau^2$ = 0.0209, $p$ = 0 | | | | |
| **Ticks–Ongoing infection (209 Studies)** | | | | |
| **Random effect meta–analysis** | 31117 | | 2.08 | [ 1.31; 2.97] |
| Heterogeneity: $I^2$ = 93.3% [92.7%; 93.9%], $\tau^2$ = 0.0242, $p$ = 0 | | | | |
| **Other animal species–Ongoing infection (10 Studies)** | | | | |
| **Random effect meta–analysis** | 810 | | 4.55 | [ 1.94; 7.93] |
| Heterogeneity: $I^2$ = 61.6% [23.5%; 80.7%], $\tau^2$ = 0.0053, $p$ = 0.0053 | | | | |
| **Other animal species–Recent infection (6 Studies)** | | | | |
| **Random effect meta–analysis** | 1627 | | 0.43 | [ 0.00; 2.92] |
| Heterogeneity: $I^2$ = 81.6% [60.6%; 91.4%], $\tau^2$ = 0.0080, $p$ < 0.0001 | | | | |
| **Other animal species–Past infection (218 Studies)** | | | | |
| **Random effect meta–analysis** | 92058 | | 12.06 | [ 9.93; 14.35] |
| Heterogeneity: $I^2$ = 98.9% [98.9%; 99.0%], $\tau^2$ = 0.0558, $p$ = 0 | | | | |

0  20  40  60  80  100

**Fig 3. Global prevalence estimate of Crimean-Congo hemorrhagic fever virus infections in humans, ticks, and other animals species.**

**Table 1. Summary of meta-analysis results for global prevalence of Crimean-Congo Hemorrhagic Fever Virus in humans, ticks, and other animal species.**

| | Prevalence. % (95% CI) | 95% Prediction interval | N Studies | N Participants | ¶H (95%CI) | §I² (95%CI) | P heterogeneity | P Egger test |
|---|---|---|---|---|---|---|---|---|
| **CCHFV case fatality rate in humans** | | | | | | | | |
| **Acute infection** | | | | | | | | |
| Overall | 11.7 [9.1–14.6] | [0.4–32.5] | 45 | 8096 | 3.3 [3–3.7] | 90.9 [88.7–92.7] | < 0.001 | 0.031 |
| Cross-sectional | 11 [8.3–14] | [0.2–31.6] | 39 | 7747 | 3.5 [3.1–3.9] | 91.6 [89.5–93.3] | < 0.001 | 0.086 |
| Low risk of bias | 13.1 [8.9–18.1] | [0–41.3] | 25 | 4912 | 4 [3.5–4.5] | 93.7 [91.8–95.2] | < 0.001 | 0.098 |
| **Recent infection** | | | | | | | | |
| Overall | 8 [1–18] | [0–50.5] | 6 | 179 | 1.9 [1.2–2.9] | 71.8 [34.7–87.8] | 0.003 | 0.001 |
| Cross-sectional | 8 [1–18] | [0–50.5] | 6 | 179 | 1.9 [1.2–2.9] | 71.8 [34.7–87.8] | 0.003 | 0.001 |
| Low risk of bias | 16 [0.9–40.5] | [0–100] | 4 | 86 | 2.1 [1.3–3.5] | 78.1 [40.9–91.9] | 0.003 | 0.032 |
| **Past infection** | | | | | | | | |
| Overall | 4.7 [0–37.7] | NA | 2 | 41 | 2.3 [1.1–4.7] | 80.8 [17.7–95.5] | 0.023 | NA |
| Cross-sectional | 4.7 [0–37.7] | NA | 2 | 41 | 2.3 [1.1–4.7] | 80.8 [17.7–95.5] | 0.023 | NA |
| Low risk of bias | 20 [0.5–51.3] | NA | 1 | 10 | NA | NA | 1 | NA |
| **CCHFV prevalence in humans** | | | | | | | | |
| **Acute infection** | | | | | | | | |
| Overall | 22.5 [15.8–30.1] | [0–87] | 67 | 35198 | 14.4 [13.9–14.9] | 99.5 [99.5–99.5] | < 0.001 | 0.001 |
| Cross-sectional | 22.1 [15–30.1] | [0–87] | 59 | 34724 | 15.3 [14.8–15.8] | 99.6 [99.5–99.6] | < 0.001 | 0.001 |
| Low risk of bias | 20.9 [11.6–31.8] | [0–84.9] | 28 | 26284 | 17 [16.2–17.8] | 99.7 [99.6–99.7] | < 0.001 | 0.006 |
| **Recent infection** | | | | | | | | |
| Overall | 11.7 [7.6–16.4] | [0–56.9] | 56 | 27173 | 9.3 [8.9–9.8] | 98.8 [98.7–99] | < 0.001 | < 0.001 |
| Cross-sectional | 12 [7.6–17.1] | [0–57.8] | 48 | 26658 | 10 [9.5–10.5] | 99 [98.9–99.1] | < 0.001 | < 0.001 |
| Low risk of bias | 19.7 [10–31.4] | [0–78.3] | 18 | 18276 | 12 [11.1–13] | 99.3 [99.2–99.4] | < 0.001 | 0.001 |
| **Past infection** | | | | | | | | |
| Overall | 4.3 [3.3–5.4] | [0–23.3] | 147 | 74900 | 6.6 [6.3–6.8] | 97.7 [97.5–97.9] | < 0.001 | 0.012 |
| Cross-sectional | 4.2 [3.2–5.3] | [0–22.9] | 135 | 73576 | 6.8 [6.5–7] | 97.8 [97.6–98] | < 0.001 | 0.018 |
| Low risk of bias | 4.4 [2.7–6.5] | [0–23.6] | 44 | 21917 | 6.3 [5.8–6.7] | 97.5 [97.1–97.8] | < 0.001 | 0.717 |
| **CCHFV prevalence in ticks** | | | | | | | | |
| **Acute exposure** | | | | | | | | |
| Overall | 2.1 [1.3–3] | [0–21.4] | 209 | 31117 | 3.9 [3.7–4] | 93.3 [92.7–93.9] | < 0.001 | 0.087 |
| Cross-sectional | 2.1 [1.3–3] | [0–21.4] | 209 | 31117 | 3.9 [3.7–4] | 93.3 [92.7–93.9] | < 0.001 | 0.087 |

*(Continued)*

**Table 1.** (Continued)

| | Prevalence. % (95% CI) | 95% Prediction interval | N Studies | N Participants | ¶H (95%CI) | §I² (95%CI) | P heterogeneity | P Egger test |
|---|---|---|---|---|---|---|---|---|
| **CCHFV prevalence in other animal species** | | | | | | | | |
| **Acute infection** | | | | | | | | |
| Overall | 4.6 [1.9–7.9] | [0–16.3] | 10 | 810 | 1.6 [1.1–2.3] | 61.6 [23.5–80.7] | 0.005 | 0.854 |
| Cross-sectional | 4.6 [1.9–7.9] | [0–16.3] | 10 | 810 | 1.6 [1.1–2.3] | 61.6 [23.5–80.7] | 0.005 | 0.854 |
| **Recent infection** | | | | | | | | |
| Overall | 0.4 [0–2.9] | [0–13.5] | 6 | 1627 | 2.3 [1.6–3.4] | 81.6 [60.6–91.4] | < 0.001 | 0.091 |
| Cross-sectional | 0.4 [0–2.9] | [0–13.5] | 6 | 1627 | 2.3 [1.6–3.4] | 81.6 [60.6–91.4] | < 0.001 | 0.091 |
| **Past infection** | | | | | | | | |
| Overall | 12.1 [9.9–14.4] | [0–54.8] | 218 | 92058 | 9.6 [9.4–9.9] | 98.9 [98.9–99] | < 0.001 | 0.001 |
| Cross-sectional | 12.1 [9.9–14.4] | [0–54.8] | 218 | 92058 | 9.6 [9.4–9.9] | 98.9 [98.9–99] | < 0.001 | 0.001 |
| Low risk of bias | 8 [4.2–12.9] | NA | 2 | 1818 | 3.5 [1.9–6.5] | 91.8 [71.5–97.6] | < 0.001 | NA |

CI: confidence interval; N: Number; 95% CI: 95% Confidence Interval; NA: not applicable.

¶H is a measure of the extent of heterogeneity, a value of H = 1 indicates homogeneity of effects and a value of H >1indicates a potential heterogeneity of effects.

§: I2 describes the proportion of total variation in study estimates that is due to heterogeneity, a value > 50% indicates presence of heterogeneity

species (Figs 2 and 3). Egger's test and funnel plot was performed to identify publication bias. The funnel plots indicate good symmetry for determining the CFR and CCHFV prevalence in other animals (S5–S14 Figs). Egger's test indicates a significant publication bias for the determination of the CFR, CCHFV prevalence in humans, ticks, and other animal species with past infections (Table 1).

## Subgroup analysis

The CCHFV CFR in humans and CCHFV prevalences observed in humans, ticks, and other animals varied widely in different regions of the world (Fig 4).

The subgroup analysis based on the qualitative variables collected is presented in S8 Table. Analysis of the data showed that CCHFV CFR were mostly reported among CCHFV suspected cases (42 studies). The pooled CFR among individuals experiencing acute CCHFV infection ranged from 30.0% [95% CI = 4.9–62.5] in patients with bleeding symptoms to 11.4% [95% CI = 8.8–14.3] in CCHFV suspected cases. The highest CFR of CCHFV acute infection in humans was recorded in probabilistic studies (p = 0.001), low-income countries (p <0.001), countries in the WHO region of Africa and South-East Asia (p <0.001), in adults (p < 0.001), and in ambulatory patients (p <0.001). The pooled CFR in recent infection ranged from 20.0% [95% CI = 0.5–51.3] in patients with bleeding symptoms to 1.9% [95% CI = 0.0–8.0] in febrile patients. The highest CFR from CCHFV recent infection in in humans was recorded in retrospective studies (p <0.001) and countries in the WHO region of Eastern Mediterranean (p <0.001). The CFR in past infection ranged from 20.0% [95% CI = 0.5–51.3] in patients with bleeding symptoms to 0.0% [95% CI = 0.0–5.4] in CCHFV suspected cases.

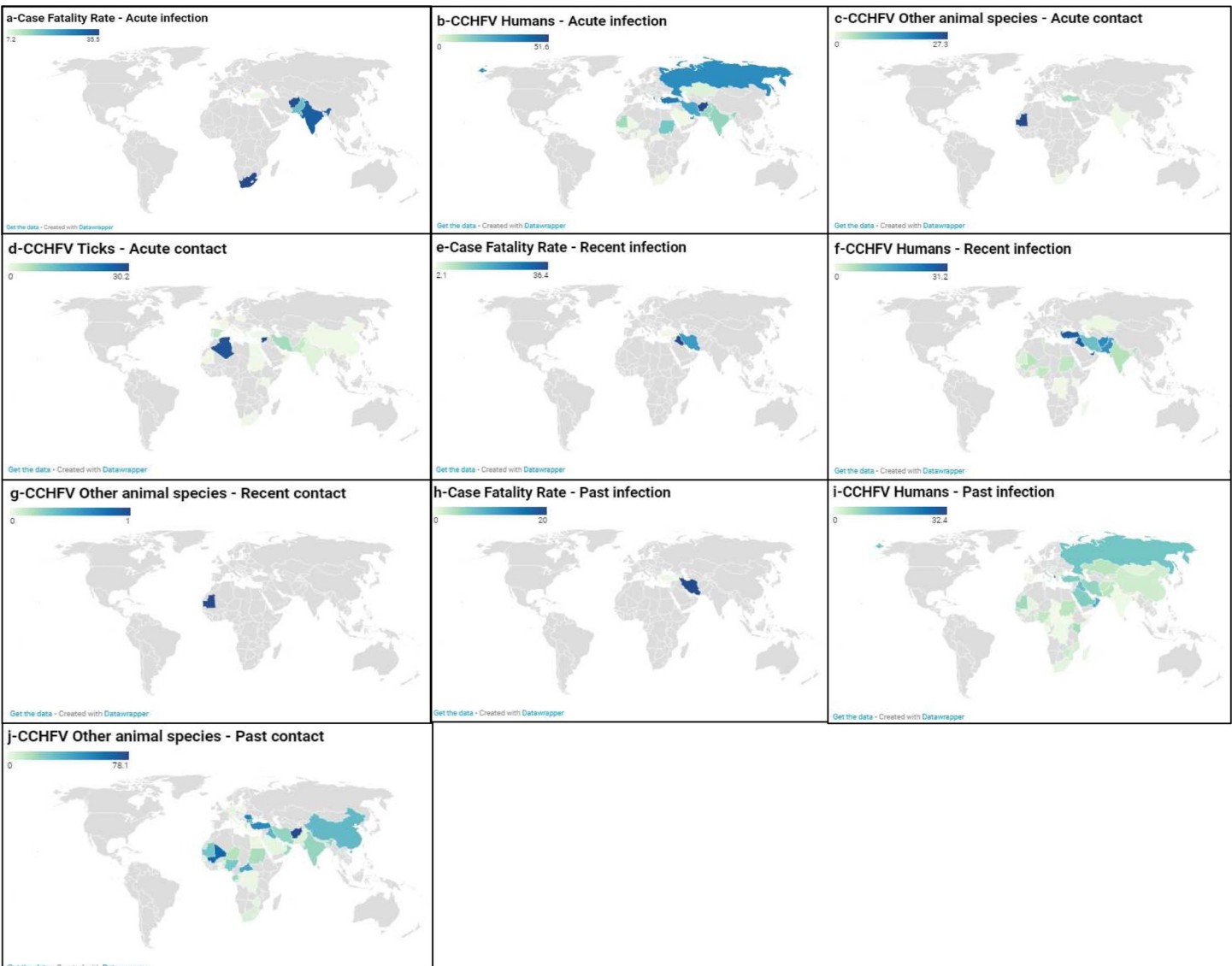

**Fig 4. Global case fatality rate, prevalence, and seroprevalence estimate of Crimean-Congo hemorrhagic fever virus in humans, ticks, and other animal species.** The letters (a, b, c, and d), (e, f, and g) and (h, I, and g) denote acute, recent and past CCHFV exposures, respectively. The letters (a, e, and h) shows the case fatality rate in humans. The letters (b, f, and, i), (c, g, and j) and (d) present the CCHFV detection rates in humans, other animal species and ticks, respectively. Map source: https://www. datawrapper.de/.

Data analysis showed that the prevalence of CCHFV was mostly reported in CCHFV suspected cases (70 studies) with acute and recent CCHFV infections as well as among apparently healthy individuals with past CCHFV infection (61 studies). The pooled prevalence of CCHFV among individuals experiencing acute infection ranged from 49.0% [95% CI = 35.3–62.8] in healthcare workers to 1.9% [95% CI = 0.4–4.2] in febrile patients. The highest prevalence of CCHFV acute infection in humans was recorded in hospital outbreaks (p <0.001), non-probabilistic studies (p < 0.001), retrospective studies (p = 0.019), upper-middle-income economies (p < 0.001), countries in the WHO region of Europe (p <0.001), and in healthcare workers and CCHFV suspected cases (p <0.001). The pooled seroprevalence of CCHFV in recent infection ranged from 31.7% [95% CI = 21.7–42.7] in CCHFV suspected cases to 0.0% [95%

CI = 0.0–0.3] in healthcare workers. The highest seroprevalence of CCHFV recent infection in humans was recorded in hospital outbreaks (p <0.001), non-probabilistic studies (p < 0.001), high-income countries (p = 0.003), countries in the WHO region of Europe and Eastern Mediterranean (p <0.001), in hospitalised patients (p <0.001), and in CCHFV suspected cases (p <0.001). The pooled seroprevalence of CCHFV in past infection ranged from 30.5% [95% CI = 22.8–38.8] in CCHFV suspected cases to 0.7% [95% CI = 0.0–1.7] in healthcare workers. The highest seroprevalence of CCHFV past infection in humans was recorded in cohort and case control studies (p < 0.001), retrospective studies (p = 0.021), upper-middle-income and high-income countries (p < 0.001), countries in the WHO region of Europe and Eastern Mediterranean (p <0.001), adults (p = 0.049), hospitalised patients (p = 0.026), and in CCHFV suspected cases (p <0.001).

In this review, CCHFV was detected in over 35 different tick species belonging to the genus: *Hyalomma*, *Rhipicephalus*, Haemaphysalis, Dermacentor, *Ixodes*, *Amblyomma*, and *Ornithodoros*. Data analysis showed that the prevalence of CCHFV was mostly reported in ticks of the genus *Hyalomma* (106 studies) and the genus *Rhipicephalus* (52 studies). Among ticks for which genus classification was possible, the pooled prevalence of CCHFV varied from 11.4% [95% CI = 0.0–63.5] in *Ornithodoros* to 0.0% [95% CI = 0.0–3.2] in *Argas*. Thirty-six studies reported positive species in pooled tested ticks. These positive species in pooled tested ticks included those of the genera *Amblyomma*, *Dermacentor*, *Haemaphysalis*, *Hyalomma*, *Ixodes* and *Rhipicephalus*. The highest prevalence of CCHFV acute infection among ticks was recorded in probabilistic studies (p = 0.008), countries in the WHO region of Eastern Mediterranean and Africa (p <0.001), and genus *Ornithodoros* and *Amblyomma* (p < 0.001).

Analysis of the data showed that the prevalence of CCHFV was mostly reported in cases of CCHFV past infection (20/23 orders). The majority of studies was based on the order *Artiodactyla* (162 studies). The pooled prevalence of CCHFV in acute infection was 6.5% [95% CI = 0.0–51.0] in *Rodentia* and 5.3% [95% CI = 3.4–7.4] in *Artiodactyla*. The highest seroprevalence of CCHFV acute infection in other animal species was recorded in prospective studies (p = 0.001), and countries in the WHO region of Africa (p = 0.004). The pooled prevalence of CCHFV in recent infection was reported only in *Artiodactyla*, 0.4% [95% CI = 0.0–2.9]. The greatest pooled seroprevalence of CCHFV in past infection was 35.6% [95% CI = 10.5–65.7] in *Perissodactyla*. The orders *Anseriformes*, *Apodiformes*, *Columbiformes*, *Eulipotyphla*, *Gruiformes*, *Hyracoidea*, *Macroscelidea*, *Pelecaniformes*, and Primates showed no evidence of CCHFV past infection. The highest seroprevalence of CCHFV past infection in other animal species was recorded in low-income and upper-middle-income countries (p <0.001), countries in the WHO region of Western Pacific (p <0.001), and in the orders of *Perissodactyla* and *Bucerotiformes* (p < 0.001).

The sources of heterogeneity were explored using univariate and multivariate metaregression and the results are shown in S9 Table. Heterogeneity in the estimate of the CFR in humans was explained at 78.0% for acute infections. The heterogeneity in the estimate of CCHFV prevalence in humans was explained at 43.9% for acute infections, 35.6% for recent infections, and 16.4% for past infections. The heterogeneity in the estimate of CCHFV prevalence in ticks was explained at 8.8% for acute infections. The heterogeneity in the estimate of CCHFV prevalence in other animal species was explained at 66.8% and 0.0% for acute and past infections respectively.

## Discussion

This study provides a summary of epidemiological data of Crimean Congo hemorrhagic fever, in humans, ticks and other animal species derived from articles published from 1974 to 2020

with studied participants recruited from 1964 to 2018. This study revealed a wide range of host species for CCHFV in ticks and other animal species. Globally, for acute infections, we estimated the CCHFV CFR of 11.7% in humans and a CCHFV prevalence of 22.5%, 4.5% and 2.1% in humans, ticks, and other animal species, respectively. For recent infections, we estimated the CCHFV CFR of 8.0% in humans and a CCHFV seroprevalence of 11.6% and 0.4% in humans and other animal species, respectively. For past infections, we estimated the CCHFV CFR of 4.7% in humans and a CCHFV seroprevalence of 4.3% and 12.0% in humans and other animal species, respectively. Highest CFR were recorded in low-income countries, Africa, South-East Asia, Eastern Mediterranean, adults, and ambulatory patients. The CCHFV prevalences were higher among healthcare workers, CCHFV suspected cases, hospitalised patients, adults, Europe, Eastern Mediterranean, upper-middle-income and high-income countries, and during hospital outbreaks. The CCHFV prevalences among ticks were higher in the genus *Ornithodoros* and *Amblyomma* and Eastern Mediterranean and Africa regions. The CCHFV prevalences in other animal species were higher in animals of the orders *Perissodactyla* and *Bucerotiformes*, Africa, Western Pacific, and low-income and upper-middle-income countries.

A study in pregnant women with CCHF from Russia, Kazakhstan and Turkey reported a case fatality rate of 34% among pregnant women and 58.5% stillbirth [31]. Another recent review by Nasirian reported an overall mean of fatality rate of 32.2% [6]. Although there is no approved antiviral for CCHFV, CCHF patient management is mainly based on supportive treatment including blood administration and providing intensive care for cases developing organ failure. Moreover, the ribavirin is encouraged in post-exposure for medical professionals [32], to prevent secondary infection or in the early stage of infection in order to reduce mortality [32, 33]. Low-income countries are known to have limited healthcare access including the above CCHF palliative CCHF case management measures [34]. This would therefore be one of the main explanatory reasons for the high case fatality rate recorded during this review in low-income countries, especially countries in the African, South-East Asian, and Eastern Mediterranean WHO regions.

In this review, high prevalences of CCHFV in humans were noted in upper-middle-income and high-income countries. Interpretation of this result should be made with caution as low-income countries have poor access to diagnosis and therefore may have and underestimated CCHFV positive numbers [34]. CCHFV geographic distribution correlates with the global distribution of the ticks which serve as vector/reservoir and are considered as crucial in maintaining endemic foci [1, 3, 9]. Although ticks of the genus *Hyalomma* are recognized as the main CCHFV vector/reservoir [9], we recorded the highest CCHFV prevalences in this review in ticks of the genus *Ornithodoros* and *Amblyomma* and in the WHO African and West Mediterranean regions. This relatively high rate of CCHFV in these tick genus suggests that more consideration should be made in these ticks because they can potentially provide a persistent virus reservoir due to their long life-span. Furthermore, the role of tick species other than *Hyalomma* is not fully understood and does need further investigation. Ticks are able to feed on various migratory birds, domestic and wild animals and play an essential role in a silent enzootic CCHFV circulation [3]. Thus, seroepidemiological studies in animals are suitable indicator to define risk area for human infections. During this review, we obtained a CCHFV pooled seroprevalence of 12.0% in animals with past infections. The data of this study show a remarkable wide range of host species of the CCHFV among animals with at least 10 animal orders and suggest that apart from small ruminants, attention should also be paid to other animal species. Investigation should be focused on animals capable of migrating, such as bats and birds, which may be infected or carry infected ticks [3, 4, 10, 35].

This study shows that the majority of studies on the CCHFV CFR and associated factors was conducted in Turkey followed by Iran, Afghanistan, Pakistan, India. It is therefore urgent to conduct more studies to estimate the CCHFV CFR and the associated factors in other regions and more particularly in low-resource settings where the burden is greater. No approved vaccine is currently available to prevent CCHFV infections. Funding agencies and public health authorities should continue to promote vaccine development programs in humans, ticks, and other animal species. Reducing human and animal exposure to ticks would be of critical importance in reducing CCHFV transmission. This could be done through the use of acaricides or the sanitation of livestock areas. Raising people's awareness of the modes of transmission and ways to prevent exposure to CCHFV would be another useful way to reduce infections. Hand hygiene and the use of personal protective equipment by healthcare workers, and more particularly during hospital outbreaks, would be effective in reducing noso-comial transmission of CCHFV. Handling of specimens from suspected CCHF cases should be carried out in specialized laboratories and by trained personnel. The prevention would also require the use of adequate protective equipment in livestock facilities, slaughterhouses, and places where animals are marketed. To expect a substantial reduction in mortality from CCHFV the application of the above measures should place special emphasis on resource-limited settings, adult, and outpatient CCHF suspected cases. Further studies on humans, ticks and animals of other species are crucial to identify new areas at risk, to monitor the spread of the virus and to ensure an adequate level of preparedness.

Although we intended to investigate most of the sources of heterogeneity in this meta-analysis, the results remain affected by a substantial unexplained residue of heterogeneity. Due to the divergent nature of the data reported in the included studies, we have incompletely explored major sources of heterogeneity such as diagnostic assays, positivity thresholds of detection assays, animal species and population categories recruited. Pooled tick analyses in some studies also compromised the ability to estimate prevalences for those studies, and could influence the estimate for ticks in this study. Another pitfall of this work is the consideration for inclusion of only article written in English or French. A search for studies in the language other than English and French in the 3 databases consulted resulted in about 200 studies to be screened for titles, abstracts and full texts for potential inclusion.

## Conclusions

Overall, our analyses revealed the presence of CCHFV worldwide in multiple human categories and a wide range of tick and animal species. This review also reveals a great variability in the CFR, prevalence and seroprevalence of CCHFV according to geographic regions, different human categories, tick species, and domestic and wild animals. These findings are important in guiding preventive actions against CCHFV.

## Supporting information

**S1 Text. Reference list of studies on Crimean-Congo hemorrhagic fever virus global case fatality rate estimate in humans.**
(PDF)

**S2 Text. Reference list of studies on Crimean-Congo hemorrhagic fever virus global prevalence in humans.**
(PDF)

**S3 Text. Reference list of studies on Crimean-Congo hemorrhagic fever virus global prevalence in ticks.**
(PDF)

**S4 Text. Reference list of studies on Crimean-Congo hemorrhagic fever virus global prevalence in other animal species.**
(PDF)

**S1 Table. Preferred reporting items for systematic reviews and meta-analyses checklist.**
(PDF)

**S2 Table. Search strategy in Medline (Pubmed).**
(PDF)

**S3 Table. Items for risk of bias assessment.**
(PDF)

**S4 Table. Main reasons of exclusion of eligible studies.**
(PDF)

**S5 Table. Risk of bias assessment.**
(PDF)

**S6 Table. Characteristics of included studies.**
(PDF)

**S7 Table. Individual characteristics of included studies.**
(PDF)

**S8 Table. Subgroup analyses of worldwide case fatality rate and prevalence of Crimean-congo hemorrhagic fever virus in humans, ticks, and other animals.**
(PDF)

**S9 Table. Univariable and multivariable meta-regression analysis on the human case fatality rate and prevalence of CCHFV in humans, ticks, and other animal species.**
(PDF)

**S1 Fig. Global case fatality rate estimate of Crimean-Congo hemorrhagic fever virus acute, recent and past infections in humans.**
(PDF)

**S2 Fig. Global prevalence estimate of Crimean-Congo hemorrhagic fever virus acute, recent and past infections in humans.**
(PDF)

**S3 Fig. Global prevalence of Crimean-congo hemorrhagic fever virus infections in ticks.**
(PDF)

**S4 Fig. Global prevalence estimate of Crimean-Congo hemorrhagic fever virus acute, recent and past exposure in other animal species.**
(PDF)

**S5 Fig. Funnel plot for publication for global case fatality rate of Crimean-congo hemorrhagic fever virus in humans with acute infections.**
(PDF)

**S6 Fig. Funnel plot for publication for global case fatality rate of Crimean-congo hemorrhagic fever virus in humans with recent infection.**
(PDF)

**S7 Fig. Funnel plot for publication for global case fatality rate of Crimean-congo hemorrhagic fever virus in humans with past infection.**
(PDF)

**S8 Fig. Funnel plot for publication for global prevalence of Crimean-congo hemorrhagic fever virus prevalence in Humans with acute infections.**
(PDF)

**S9 Fig. Funnel plot for publication for global prevalence of Crimean-congo hemorrhagic fever virus prevalence in Humans with recent infections.**
(PDF)

**S10 Fig. Funnel plot for publication for global prevalence of Crimean-congo hemorrhagic fever virus prevalence in Humans with past infections.**
(PDF)

**S11 Fig. Funnel plot for publication for global prevalence of Crimean-congo hemorrhagic fever virus in ticks.**
(PDF)

**S12 Fig. Funnel plot for publication for global prevalence of Crimean-congo hemorrhagic fever virus in other animals with acute infections.**
(PDF)

**S13 Fig. Funnel plot for publication for global prevalence of Crimean-congo hemorrhagic fever virus in other animals recent infections.**
(PDF)

**S14 Fig. Funnel plot for publication for global prevalence of Crimean-congo hemorrhagic fever virus in other animals with past infections.**
(PDF)

## Author Contributions

**Conceptualization:** Jean Thierry Ebogo Belobo, Sebastien Kenmoe, Richard Njouom.

**Data curation:** Jean Thierry Ebogo Belobo, Sebastien Kenmoe, Cyprien Kengne-Nde, Cynthia Paola Demeni Emoh, Arnol Bowo-Ngandji, Serges Tchatchouang, Jocelyne Noel Sowe Wobessi, Chris Andre Mbongue Mikangue, Hervé Raoul Tazokong, Sandrine Rachel Kingue Bebey, Efietngab Atembeh Noura, Aude Christelle Ka'e, Raïssa Estelle Guiamdjo Simo, Abdou Fatawou Modiyinji, Dimitri Tchami Ngongang, Emmanuel Che, Sorel Kenfack, Nathalie Diane Nzukui, Nathalie Amvongo Adjia, Isabelle Tatiana Babassagana, Gadji Mahamat, Donatien Serge Mbaga.

**Formal analysis:** Sebastien Kenmoe, Cyprien Kengne-Nde.

**Methodology:** Jean Thierry Ebogo Belobo, Sebastien Kenmoe, Cyprien Kengne-Nde, Cynthia Paola Demeni Emoh, Arnol Bowo-Ngandji, Serges Tchatchouang, Jocelyne Noel Sowe Wobessi, Chris Andre Mbongue Mikangue, Hervé Raoul Tazokong, Sandrine Rachel Kingue Bebey, Efietngab Atembeh Noura, Aude Christelle Ka'e, Raïssa Estelle Guiamdjo Simo, Abdou Fatawou Modiyinji, Dimitri Tchami Ngongang, Emmanuel Che, Sorel

Kenfack, Nathalie Diane Nzukui, Nathalie Amvongo Adjia, Isabelle Tatiana Babassagana, Gadji Mahamat, Donatien Serge Mbaga, Wilfred Fon Mbacham, Serge Alain Sadeuh-Mbah.

**Project administration:** Sebastien Kenmoe, Richard Njouom.

**Supervision:** Sebastien Kenmoe, Wilfred Fon Mbacham, Serge Alain Sadeuh-Mbah, Richard Njouom.

**Validation:** Jean Thierry Ebogo Belobo, Sebastien Kenmoe, Cyprien Kengne-Nde, Cynthia Paola Demeni Emoh, Arnol Bowo-Ngandji, Serges Tchatchouang, Jocelyne Noel Sowe Wobessi, Chris Andre Mbongue Mikangue, Hervé Raoul Tazokong, Sandrine Rachel Kingue Bebey, Efietngab Atembeh Noura, Aude Christelle Ka'e, Raïssa Estelle Guiamdjo Simo, Abdou Fatawou Modiyinji, Dimitri Tchami Ngongang, Emmanuel Che, Sorel Kenfack, Nathalie Diane Nzukui, Nathalie Amvongo Adjia, Isabelle Tatiana Babassagana, Gadji Mahamat, Donatien Serge Mbaga, Wilfred Fon Mbacham, Serge Alain Sadeuh-Mbah, Richard Njouom.

**Writing – original draft:** Jean Thierry Ebogo Belobo, Sebastien Kenmoe.

**Writing – review & editing:** Jean Thierry Ebogo Belobo, Sebastien Kenmoe, Cyprien Kengne-Nde, Cynthia Paola Demeni Emoh, Arnol Bowo-Ngandji, Serges Tchatchouang, Jocelyne Noel Sowe Wobessi, Chris Andre Mbongue Mikangue, Hervé Raoul Tazokong, Sandrine Rachel Kingue Bebey, Efietngab Atembeh Noura, Aude Christelle Ka'e, Raïssa Estelle Guiamdjo Simo, Abdou Fatawou Modiyinji, Dimitri Tchami Ngongang, Emmanuel Che, Sorel Kenfack, Nathalie Diane Nzukui, Nathalie Amvongo Adjia, Isabelle Tatiana Babassagana, Gadji Mahamat, Donatien Serge Mbaga, Wilfred Fon Mbacham, Serge Alain Sadeuh-Mbah, Richard Njouom.

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
