## [Decision Letter · Decision Letter 0]

20 Dec 2020

Dear PhD Njouom,

Thank you very much for submitting your manuscript "Worldwide epidemiology of Crimean-Congo Hemorrhagic Fever Virus in humans, ticks and other animal species, a systematic review and meta-analysis." for consideration at PLOS Neglected Tropical Diseases. As with all papers reviewed by the journal, your manuscript was reviewed by members of the editorial board and by several independent reviewers. In light of the reviews (below this email), we would like to invite the resubmission of a significantly-revised version that takes into account the reviewers' comments. 

Reviewers provided a range of opinions regarding your submission. Please consider making significant revisions to your document to address these concerns. The authors are also encouraged to move some of their smaller figures and tables to the main text where they could provide additional value and clarity to the manuscript. In addition, careful copy-editing must be performed as the PLoS editorial team does not edit manuscripts.

We cannot make any decision about publication until we have seen the revised manuscript and your response to the reviewers' comments. Your revised manuscript is also likely to be sent to reviewers for further evaluation.

Sincerely,

Michael R Holbrook, PhD

Associate Editor

Jeremy Camp, PhD

Deputy Editor

Reviewers provided a range of opinions regarding your submission. Please consider making significant revisions to your document to address these concerns. The authors are also encouraged to move some of their smaller figures and tables to the main text where they could provide additional value and clarity to the manuscript. In addition, careful copy-editing must be performed as the PLoS editorial team does not edit manuscripts.

Reviewer's Responses to Questions

**Key Review Criteria Required for Acceptance?**

**Methods**

-Are the objectives of the study clearly articulated with a clear testable hypothesis stated?

-Is the study design appropriate to address the stated objectives?

-Is the population clearly described and appropriate for the hypothesis being tested?

-Is the sample size sufficient to ensure adequate power to address the hypothesis being tested?

-Were correct statistical analysis used to support conclusions?

-Are there concerns about ethical or regulatory requirements being met?

Reviewer #1: -Are the objectives of the study clearly articulated with a clear testable hypothesis stated?

In my view, the research question of the study, “to assess CCHFV worldwide prevalence and seroprevalence in wildlife, livestock, ticks and humans” makes no sense. Various factors have an impact on the incidence of CCHF in different regions. These factors may vary considerably from region to region. This is not sufficiently reflected in the presented research. 

-Is the study design appropriate to address the stated objectives?

No, because the objective is not meaningful.

-Is the population clearly described and appropriate for the hypothesis being tested?

Not applicable

-Is the sample size sufficient to ensure adequate power to address the hypothesis being tested?

Not applicable

-Were correct statistical analysis used to support conclusions?

The methodology is not always transparent (e.g. the start date of the search: Methods: No date mentioned; Discussion (line 364: “articles published from 1974”; the screening procedure [e. g. lines 135-136: “All the articles identified were first reviewed based on the title and abstract …, and afterwards they were reviewed base on the full text; lines 171-172 suggests that articles were excluded after reading titles and abstracts; “metaregression” (line 162) and “sensitivity analyses” are mentioned, but not described in sufficient detail).

-Are there concerns about ethical or regulatory requirements being met?

No

Reviewer #2: Very well thought out.

Reviewer #3: The manucript is a systematic review and meta-analysis on Crimean-Congo hemorrhagic fever virus (CCHFV), with emphasis on global epidemiology. The search and evaluation methods are valid and well-explained. 

However, the manuscript definitely requires major edits in form, that are explained below: 

- The claims of originality indicated in several statements is unnecessary and do not reflect the content. The authors seem to emphasize case fatality rate for this purpose in many instances in the text, summary, keywords etc., to set this effort apart from other recent reviews on CCHFV (please consider how contradictory the statement in the the very first lines of the introduction (line 55) appears in this regard). Comparison of prevalence rates in previous meta-analyses (as observed in Discussion) is also unnecassary and can be due to differences in methodology.

 - In the text, many statements suffer from awkward phrasing with occasional grammatical errors. A language review by a native speaker is required.

- Table 1 is hard to follow and is unlikely to fit in a print format. 

- Although obviously cited in this format, the references are practically too many. This reviewer wonders whether it is possible to transfer the references to a supplement format. The citations in the text (such as in lines 244-245, 250-251) are impossible to follow. 

- Figures 2 and 3 are hard to follow, maybe due to the text format in the document without titles. The figures except for Fig.1 are practically tables and can be organized as such. 

 - The format in Fig.6 also prevents understanding of the color-based differences in prevalence etc. 

- There are also occational typing errors (Line 148: “Hoy et al.”; line 162: “meta-regression”), which are minor considering the issues outlined above.

**Results**

-Does the analysis presented match the analysis plan?

-Are the results clearly and completely presented?

-Are the figures (Tables, Images) of sufficient quality for clarity?

Reviewer #1: -Does the analysis presented match the analysis plan?

Yes

 -Are the results clearly and completely presented?

Yes.

 -Are the figures (Tables, Images) of sufficient quality for clarity?

Yes.

Reviewer #2: Yes

Reviewer #3: Results are relatively hard to follow due to the problems outlined above.

**Conclusions**

-Are the conclusions supported by the data presented?

-Are the limitations of analysis clearly described?

-Do the authors discuss how these data can be helpful to advance our understanding of the topic under study?

-Is public health relevance addressed?

Reviewer #1: -Are the conclusions supported by the data presented?

No

 -Are the limitations of analysis clearly described?

No, a critical appraisal of the chosen methods is missing.

 -Do the authors discuss how these data can be helpful to advance our understanding of the topic under study?

No, not sufficiently.

 -Is public health relevance addressed?

No, not sufficiently.

Reviewer #2: Yes

Reviewer #3: (No Response)

**Editorial and Data Presentation Modifications?**

Reviewer #1: (No Response)

Reviewer #2: (No Response)

Reviewer #3: (No Response)

**Summary and General Comments**

Reviewer #1: (No Response)

Reviewer #2: The article by Thierry et al. describes a very thorough analysis on data from peer-reviewed papers on various aspects of CCHF/CCHFV biology. It is extremely well written and the results improve the fields clarity on these subjects. I have only minor suggestions:

1 Line 57: It s incorrect to state that CCFV circulates worldwide. As far as I can tell, and judging by the maps presented, it is not in the New World. Please re-word. 

2. Line 91: Same as above

In assessment of study quality, please specify the total number of papers for each section (for example see line 182). Add the n size for papers looking at individual ticks, pooled ticks and other animals

3 Can the authors add a table listing "other" animals studied to date and a part of this analysis

Reviewer #3: In conclusion, although appears to be a result of significant work and has much to say, the manuscript in its current form is lacking the organization and impact expected from such a work. Considering that the required edits, only basically outlined in this review will practically result in entire rewriting, I recommend a new submission and will be available for review. However, major editing with correction of the above-mentioned issues is also acceptable, leaving the final decision to the editor.

PLOS authors have the option to publish the peer review history of their article (what does this mean?). If published, this will include your full peer review and any attached files.

Reviewer #1: No

Reviewer #2: No

Reviewer #3: No
---

## [Decision Letter · Decision Letter 1]

15 Feb 2021

Dear PhD Njouom,

Thank you very much for submitting your manuscript "Worldwide epidemiology of Crimean-Congo Hemorrhagic Fever Virus in humans, ticks and other animal species, a systematic review and meta-analysis." for consideration at PLOS Neglected Tropical Diseases. As with all papers reviewed by the journal, your manuscript was reviewed by members of the editorial board and by several independent reviewers. The reviewers appreciated the attention to an important topic. Based on the reviews, we are likely to accept this manuscript for publication, providing that you modify the manuscript according to the review recommendations. 

Sincerely,

Michael R Holbrook, PhD

Associate Editor

Jeremy Camp

Deputy Editor

Reviewer's Responses to Questions

**Key Review Criteria Required for Acceptance?**

**Methods**

-Are the objectives of the study clearly articulated with a clear testable hypothesis stated?

-Is the study design appropriate to address the stated objectives?

-Is the population clearly described and appropriate for the hypothesis being tested?

-Is the sample size sufficient to ensure adequate power to address the hypothesis being tested?

-Were correct statistical analysis used to support conclusions?

-Are there concerns about ethical or regulatory requirements being met?

Reviewer #2: (No Response)

Reviewer #4: This systematic review of CCHFV prevalence/seroprevalence sets itself apart from others by focussing on four categories: wildlife, livestock, ticks, and humans. The study design follows PRISMA guidelines and has been used in other published studies, with inclusion criteria clearly laid out. The quantitative (and some qualitative) methodologies are appropriate for this type of meta-analysis, with due consideration of potential sources of bias.

Minor comment: I understand the need to filter out articles written in any language other than English or French for translation purposes, but it would be useful to know if this constituted a large or small number of articles. How many were in other languages?

Reviewer #5: the methodology is adequately described

**Results**

-Does the analysis presented match the analysis plan?

-Are the results clearly and completely presented?

-Are the figures (Tables, Images) of sufficient quality for clarity?

Reviewer #2: (No Response)

Reviewer #4: 312 studies ultimately contributed to 802 sources of data in the four categories of interest (wildlife, livestock, ticks, humans). Table 1 is well-presented and shows the summary of meta-analysis results. It may be helpful to have an additional table before this one which summarizes the baseline characteristics, which are currently only laid out in the text. Some of the maps in Figure 4 are not given detailed enough titles. This could be helped with labels for each map (a, b, c, etc.), alongside a more details caption for the figure which lays out exactly what each map shows.

Reviewer #5: The results are comprehensively presented

**Conclusions**

-Are the conclusions supported by the data presented?

-Are the limitations of analysis clearly described?

-Do the authors discuss how these data can be helpful to advance our understanding of the topic under study?

-Is public health relevance addressed?

Reviewer #2: (No Response)

Reviewer #4: The conclusions are well supported by the results presented, with limitations and public health implications explicitly addressed, at least in a general sense. It would be helpful to have specific public health implications laid out by world region, according to the geographic variation seen in the maps in Figure 4. 

The authors note that caution should be given to the finding that high prevalence of CCHFV were not noted in low-income countries due to poor access to diagnostic resources. Are there any further analyses that may be done that might be able to model/quantify this discrepancy and perhaps estimate the degree of underreporting based on diagnostic capacity? A discussion around this point would be helpful and perhaps guide future studies.

Reviewer #5: The authors need to provide further discussion to clarify some of their findings. (see general comments below)

**Editorial and Data Presentation Modifications?**

Reviewer #2: (No Response)

Reviewer #4: Minor comment: no need to capitalise “Domestic and Wild Animals” (line 87)

Reviewer #5: The use of English could be improved.

**Summary and General Comments**

Reviewer #2: The authors have adequately addressed my suggestions.

Reviewer #4: Overall, this is a very clearly written and thoughtful review/meta-analysis which creates further knowledge about global variations in CCHFV prevalence and seroprevalence in multiple hosts/vectors.

Reviewer #5: The authors have submitted a comprehensive meta analysis and systematic review of CCHF literature on the epidemiology of CCHFV in human, tick and other animals. The methodology is clearly described but there are some areas that the authors may want to provide clarification for the reader.

1.perhaps the authors could consider replacing "ongoing" with "acute". Acute infections is more commonly used to describe CCHFV infections detected using virus, antigen RNA etc

2.line 244, perhaps define what is included in sero-prevalence, eg did you take data from articles describing IgG, IgM and/or both IgG and IgM? 

3. In line 218 the authors indicate that studies mostly included suspected cases and febrile patients, based on this comment, do the participant numbers described in the results include suspected and confirmed cases, for example the overall sero-prevalence in 74 900 participants with past infections was 4.3%? perhaps this could be clarified for all the results provided for humans and other animals. 

4. line 246, is this seroprevalence you refer to?

5. The analysis of tick data showed prevalence was mostly reported in Hyalomma and Rhipicephalus however in line 325 the authors refer to results from pooled prevalence and comment that the highest prevalence was recorded in the genera Ornithodoros and Amblyomma. this was also highlighted in the abstract and author summary. although CCHFV has been isolated from more than 30 species, the role of tick species other than Hyalomma is not fully understood and does need confirmation. these results need to be discussed further (line 395) and clarified in the context of the current knowledge of known vector species to prevent misinterpretation.

Overall the authors have submitted a comprehensive and useful review that will be strengthened by additional discussion in the context of current knowledge on CCHF.

PLOS authors have the option to publish the peer review history of their article (what does this mean?). If published, this will include your full peer review and any attached files.

Reviewer #2: No

Reviewer #4: No

Reviewer #5: No

Figure Files:

Data Requirements:

Reproducibility:

References

---

## [Editor Report · Decision Letter 2]

8 Mar 2021

Dear PhD Njouom,

We are pleased to inform you that your manuscript 'Worldwide epidemiology of Crimean-Congo Hemorrhagic Fever Virus in humans, ticks and other animal species, a systematic review and meta-analysis.' has been provisionally accepted for publication in PLOS Neglected Tropical Diseases.

The authors have adequately addressed reviewer concerns. However, it is recommended that the authors carefully proof their manuscript when it is returned as there are some typos throughout and PLoS staff do not copy-edit. Most importantly, please ensure that all Genera and Families are appropriately italicized. In several instances they are not. Note that there are differences in typography of taxonomic names between the ICZN and the ICTV (taxa higher than genus need not be italicized for zoological names whereas all virological taxa should be italicized). Also:

Line 54: should be "to the"

Line 287: There are two "."

Line 389: small "i"

Line 437: should be "genera" not "genus"

Best regards,

Michael R Holbrook, PhD

Associate Editor

Jeremy Camp

Deputy Editor

---

## [Editor Report · Acceptance letter]

15 Apr 2021

Dear PhD Njouom,

We are delighted to inform you that your manuscript, "Worldwide epidemiology of Crimean-Congo Hemorrhagic Fever Virus in humans, ticks and other animal species, a systematic review and meta-analysis.," has been formally accepted for publication in PLOS Neglected Tropical Diseases.

Best regards,

Shaden Kamhawi

co-Editor-in-Chief

Paul Brindley

co-Editor-in-Chief
